# Hybrid Data-Driven Deep Learning Framework for Material Mechanical Properties Prediction with the Focus on Dual-Phase Steel Microstructures

**DOI:** 10.3390/ma16010447

**Published:** 2023-01-03

**Authors:** Ali Cheloee Darabi, Shima Rastgordani, Mohammadreza Khoshbin, Vinzenz Guski, Siegfried Schmauder

**Affiliations:** 1Institute for Materials Testing, Materials Science and Strength of Materials, University of Stuttgart, Pfaffenwaldring 32, 70569 Stuttgart, Germany; 2Department of Mechanical Engineering, Shahid Rajaee Teacher Training University, Lavizan, Tehran 1678815811, Iran

**Keywords:** deep learning, material properties, dual-phase steel, micromechanical modeling, phase field simulation

## Abstract

A comprehensive approach to understand the mechanical behavior of materials involves costly and time-consuming experiments. Recent advances in machine learning and in the field of computational material science could significantly reduce the need for experiments by enabling the prediction of a material’s mechanical behavior. In this paper, a reliable data pipeline consisting of experimentally validated phase field simulations and finite element analysis was created to generate a dataset of dual-phase steel microstructures and mechanical behaviors under different heat treatment conditions. Afterwards, a deep learning-based method was presented, which was the hybridization of two well-known transfer-learning approaches, ResNet50 and VGG16. Hyper parameter optimization (HPO) and fine-tuning were also implemented to train and boost both methods for the hybrid network. By fusing the hybrid model and the feature extractor, the dual-phase steels’ yield stress, ultimate stress, and fracture strain under new treatment conditions were predicted with an error of less than 1%.

## 1. Introduction

Dual-phase (DP) steels are a family of high-strength low-alloy steels that exhibit high strength and good formability. They have, therefore, found extensive use in the automotive industry [1]. Their promising properties can be attributed to their microstructure, which consists of hard martensite islands and a soft ferrite matrix. This microstructure leads to high formability, continuous yielding behavior, high strength, high strain hardening rate, and low yield stress-to-tensile strength ratio [2].

One of the fundamental objectives of materials science and engineering is the development of reliable working models which connect process parameters, microstructures, and material properties. Many models have been developed for analyzing each individual domain. For example, phase field modeling (PFM) can simulate the phase transformations during heat treatment [3,4], and finite element analysis (FEA), can be used to obtain the mechanical response of a microstructure [5]. These have also been combined [6,7]. This generally takes the form of a PFM analysis obtaining a representative volume element (RVE) of a multiphase material that has undergone heat treatment and the resulting microstructure being loaded with specific boundary conditions to obtain its fracture stress using FEA.

These models have an inherent deficiency that they only work on a limited part of the problem and connecting all the effects can be very challenging. Furthermore, they can only be used to analyze a particular configuration after it has been conceived. They do not have any predictive power and must be run many times to obtain a suitable model. Currently, using these approaches for designing new materials is very costly, time-consuming, and requires substantial lab work.

These problems can be avoided by assigning the modern advancements of machine learning methods [8]. Machine learning and deep learning, and especially their subcategories, such as artificial neural networks (ANN) and convolutional neural networks (CNN), are being introduced in materials science and engineering because they can accelerate the processes and, in some cases, reduce the required physical experiments [9,10,11]. These models can also automate different layers of material characteristic investigations [12]. Different scaled microstructure studies, from macro and continuum levels to the atomic and micro scales, could benefit from the recent developments in ANN techniques [13,14]. Additionally, methods such as phase field modeling could assist the researchers in 2D and 3D simulations, enhancing the dataset for further steps to an ANN model [15,16]. These new tools make the final aim of tailoring the material features achievable and within reach.

The classic paradigm of microstructural behavior studies needs to be revised. Recent material informatics developments could magnify machine learning approaches’ vital role in quantitative microstructural subjects [17,18,19]. Thus, the need to expand the knowledge of neural network applications in materials science and engineering is evident. In the last decade, various methods have been implemented to predict the characteristics of different materials [20].

This work represents a timely, advanced computational methodology with a wide range of implications to help the materials community [21]. The novelty of this work is twofold: we use validate and utilize simulations of heat treatment to generate microstructures, which reduces the cost associated with creating a machine learning dataset. Additionally, we introduce a hybrid machine learning model and apply it to a materials science problem. In the first step of this study, since having an extensive data set for training is the prerequisite of a deep neural network, about 1000 images were generated with a phase field model. About 10 percent of the whole data set was randomly chosen for the testing set. For this study, different models, including simple CNN and transfer learning methods, were investigated and two algorithms with faster optimization behavior, VGG16 [22] and ResNet [23], were paralleled and named “Hybrid Model”. Not every model showed promising results regarding the prediction of tensile stress and fracture strain. However, with an error of less than 1% for the prediction of ultimate stress and yield stress for the testing data, and about 0.5% for the training set, this model could respond ideally. This fast and accurate technique could be applied to different alloy data sets, giving scientists a better overview of the metal characteristics. 

## 2. Data Generation

### 2.1. Overview

In this study, a large number of phase field (PF) heat treatment simulations were performed to generate artificial DP steel microstructures. These microstructures were then analyzed using finite element analysis (FEA) to obtain the mechanical response of those steels. Consequently, a dataset containing process parameters, resulting microstructure, and mechanical properties was created, which was then used in Section 3 to train a machine learning system. A high-level illustration of the process is shown in Figure 1.

The following sections describe and validate the PF and FEA models and then explain how the two data pipelines work together to create the final dataset.

### 2.2. Multiphase Field Simulation

#### 2.2.1. Basic Theory

The phase field equation was implemented by Steinbach et al. [24,25] to predict microstructure evolution. In this approach, a phase field parameter φ is defined for each phase, which changes between 0 and 1 during the process. Parameter $\varphi_{\alpha }$ indicates the local fraction of phase (α) in a grain, which means the sum of the local fraction of phases is equal 1 ($\sum \varphi_{\alpha }=1$). In this paper, MICRESS^®^ software, version 7, was used for the phase field simulation, and the rate of parameter φ during the process is shown as Equation (1) [26]:(1)$${\dot{\varphi }}_{\alpha }=\sum_{\alpha \neq \beta }^{n}M_{\alpha \beta }^{\varphi }[b\Delta G_{\alpha \beta }-\sigma_{\alpha \beta }\left(K_{\alpha \beta }+A_{\alpha \beta }\right)+\sum_{\alpha \neq \beta \neq \gamma }^{\upsilon }j_{\alpha \beta \gamma }],$$
where the parameters α, β, and γ show the different phases, and n is the number of phases in the simulation. Parameter $M_{\alpha \beta }^{\varphi }$, given as Equation (2), is related to the interface mobility between phases α and β, which is a function of the kinetic coefficient in the Gibbs–Thomson equation: (2)$$M_{\alpha \beta }^{\varphi }=\frac{\mu_{\alpha \beta }^{G}}{1+\frac{\mu_{\alpha \beta }^{G}\eta \Delta s_{\alpha \beta }}{8}\left\{\sum_{i}m_{i}^{l}\sum_{i}\left[{\left(D_{\alpha }^{\mathrm{ij}}\right)}^{-1}\left(1-k_{j}\right)c_{j\alpha }\right]\right\}},$$
where η and $\Delta s_{\alpha \beta }$ are the thickness of the interface and entropy of fusion between the phases, respectively. Additionally, the parameters $m_{i}^{l}$ and $D_{\alpha }^{ij}$ represent the liquidus line slop for component i and the diffusion matrix, respectively, and $k_{j}$ is related to the partition coefficient. 

The expression inside the brackets represents the required force for moving the interface between phases α and β. Parameter b is a pre-factor and is calculated using Equation (3). The parameters $\Delta G_{\alpha \beta }$ and $K_{\alpha \beta }$ show the difference in Gibbs energy and pairwise curvature between the two phases, as indicated in Equations (4) and (5), respectively. $J_{\alpha \beta \gamma }$ is related to the triple junction between three phases through Equation (6):(3)$$b=\frac{\pi }{\eta }\left(\varphi_{\alpha }+\varphi_{\beta }\right)\sqrt{\varphi_{\alpha }\varphi_{\beta }},$$
(4)$$\Delta G_{\alpha \beta }=\frac{1}{\nu^{m}}\left(\mu_{\beta }^{0}-\mu_{\alpha }^{0}\right),$$
(5)$$K_{\alpha \beta }=\frac{\pi^{2}}{2\eta^{2}}\left(\varphi_{\beta }-\varphi_{\alpha }\right)+\frac{1}{2}\left(\nabla^{2}\varphi_{\beta }-\nabla^{2}\varphi_{\alpha }\right),$$
(6)$$J_{\alpha \beta \gamma }=\frac{1}{2}\left(\sigma_{\beta \gamma }-\sigma_{\alpha \gamma }\right)\left(\frac{\pi^{2}}{\eta^{2}}\varphi_{\gamma }+\nabla^{2}\varphi_{\gamma }\right).$$

#### 2.2.2. Validation of PF Simulations

Before using the PF model for generating microstructures under different heat treatment conditions, the model’s accuracy for simulating the basic heat treatment must be validated against experiments. Here, the step quenching heat treatment process routine for the production of DP steel from low carbon steel, shown in Figure 2, is simulated using phase field simulation in MICRESS software. Afterwards, the same heat treatment procedure is also carried out experimentally, and the resulting microstructures are compared.

The base material used in the PF simulations was a ferritic–pearlitic steel with the chemical composition given in Table 1. To reduce computational costs, the heat treatment simulations started from the fully austenitic microstructure and the morphology for this state was calculated using MatCalc software, version 6.03 [27]. Afterwards, the step quenching heat treatment was simulated, resulting in the formation of ferrite and martensite phases. It was assumed that the remaining austenite phase is wholly transformed into martensite below the martensite temperature. Additionally, based on the chemical composition given in Table 1 and using the equation in the study [28], the martensite starting temperature ($M_{s}$) was calculated to be 417.34 °C. For this particular heat treatment based on the step quenching shown in Figure 2, first the fully austenitic microstructure was cooled from 1100 °C to 770 °C, then held for 10 min, and finally quenched in water to room temperature.

In this study, a binary phase diagram was implemented for the simulation. Table 2 provides information on carbon and magnesium concentration and proportionality driving pressure ($L_{ij}$) at $T_{1}$, which were calculated using MatCalc. Some other phase interaction properties, such as interface kinetic coefficient and interface mobility, were extracted from the literature and are shown in Table 3. For carbon’s diffusion properties, the maximal diffusion coefficient ($D_{0}$) in ferrite and austenite were set to $2.20\times 10^{-4}$ and 0.15×10−4 m2s; and the activation energy for diffusion (Q) were set to 122.5 and 142.1 KJmol, respectively [29,30,31]. The diffusion of magnesium was ignored in this study and the “phase concentration” model in MICRESS and periodic boundary conditions (named PPPP in MICRESS) were used. Figure 3a–e illustrates the sample progression of the heat treatment.

The only output taken from the PF models for the FEA is the final microstructure geometry. This means that to validate the PF models, it is only necessary to make sure they predict martensite volume fraction, average phase size, and morphology (banded or equiaxed) correctly. Figure 3e,f shows the simulated and experimental microstructures resulting from the described heat treatment. There is a good agreement between the results, as both microstructures have the same martensite volume fraction (34%), average phase size (15 μm) and morphology (equiaxed). This means that the utilized multiphase model can accurately predict the experimental results. Therefore, this validated model is used for simulating the final microstructure after undergoing heat treatment under different conditions.

### 2.3. FEM Simulation

#### 2.3.1. FEA Parameters

This section describes the process of creating, analyzing, and validating micromechanical FEA models based on the PF simulations. After a microstructure is generated using PFM, it can be used as a representative volume element (RVE). The parameters for a single simulation are explained here, and the next section explains how a large number of simulations is performed.

The material properties of the ferrite and martensite phases are essential factors to consider. It is well known that they change with the process parameters [33,34], but to simplify the process, the flow curves were taken from DP600 steel, as shown in Figure 4, which was reported in a previous study [2]. Damage in the martensite phase was ignored, and the Johnson–Cook damage model was used for the ferrite phase. Since the test was executed at room temperature with constant strain rates, $D_{4}$ and $D_{5}$ were ignored and local fracture strain under uniaxial loading was predicted by [2] to be 0.4. Finally, $D_{1}$, $D_{2}$, and $D_{3}$ were found to be 0.17, 0.80, and −0.7, respectively. Darabi et al. [35] showed that there is no difference in stress–strain curves of RVEs loaded under periodic and symmetric boundary conditions. Therefore, symmetric (linear displacement) boundary conditions were applied to the RVE. 

#### 2.3.2. Validation of the FE Simulation

The main outputs of the analysis were the yield strength, UTS, and fracture points. To obtain the mentioned properties, the model’s equivalent plastic strain and von Mises stress were homogenized using the method described in our previous work [2] to obtain the stress–strain curve. Afterward, the model’s Young’s modulus was calculated based on the 2% offset method, and finally, the yield strength, UTS, and fracture points were found based on the curve.

Table 4 compares the experimental and numerical results, showing that the numerical model can predict the mechanical behavior of the simulated microstructure. Therefore, this micromechanical model can predict the mechanical behavior of microstructures generated using PF simulations.

### 2.4. Data Pipelines

The main goal of PFM and FEA is to generate a large amount of reliable data for training and testing the machine learning models. Since the model parameters are determined and the validity of the models is examined, we can automate the process for each of the PFM and FEA data pipelines and connect them together to create the full dataset.

#### 2.4.1. PFM Data Pipeline

The PFM parameters are based on the various points in Figure 2. Table 5 shows them with a short description and selected values. There is a total of four variable heat treatment parameters, which result in 1188 different data points. To automate such a large number of PFM analyses, a base MICRESS^®^ driving (.dri) file was created and extensively tested. Afterwards, scripts were written that changed the parameters and saved new .dri files. Additionally, the PFM process was divided into two steps to reduce computational time. The first step was heat treatment, which was until the time $t_{2}$ in Figure 2 was reached, and the second step restarted the PFM analysis and quenched the microstructure. This procedure greatly reduces computational time because, although the second step had to be performed 1188 times, the first step was performed only 396 times. In the end, the microstructures were saved as VTK files, which were used as input for creating FEA models in Section 2.4.2. They were also saved as images to be directly used by the machine learning model. The PFM data pipeline is shown in the red section of Figure 1.

#### 2.4.2. FEA Data Pipeline

MICRESS^®^ PFM software allows output to a number of different file formats. To enable the easy creation of FEA models, output is requested in The Visualization Toolkit (VTK) file format, which can be read using readily available software libraries. The output used for modeling was the “phas” variable, which shows the phase of each element, i.e., each element was either ferrite, martensite, or part of the phase interface. Interface elements were converted to ferrite in subsequent operations.

A Python script was written that extracted the phase distribution from the VTK file and passed it to the open-source VCAMS library, which created an Abaqus^®^ input file containing the elements with the proper phase labels, as well as linear displacement boundary conditions. Another script was written for the Abaqus Python environment that opened the input file and defined the rest of the simulation parameters, such as the material assignment, etc. described in Section 2.3.1.

The main script then submitted the analysis and passed the final ODB to an Abaqus Python script that post-processed the results. This included homogenization of stress and strains using the method described in Ref. [2] to obtain the stress–strain curve, determine the elastic modulus based on the 2% offset method, and find the yield strength, UTS, in addition to fracture strains. These were then written to a file that mapped each model with its output. Pictures of the microstructure and the stress–strain curve were also saved so they can be audited if necessary. The FEA data pipeline is illustrated in the blue section of Figure 1.

## 3. Deep Learning Approaches

### 3.1. Introduction and Overview

Inspired by brain biology, artificial neural networks (ANNs) allow for the modeling of complex patterns. Nowadays, various methods have also been applied to compare the different performances of AN networks in computational mechanics [36,37,38]. The attention paid to ANN in recent years has led to the flourishing of methods such as transfer learning, which allows for loading new datasets onto pre-trained models, greatly reducing the effort required for training the neural network [39].

In order to design safe and functional parts, we need information about the material’s mechanical properties, such as ultimate stress (UTS), stiffness (E), yield stress (Y), fracture strain (F), elongation, fatigue life, etc. The same properties are also expected when we are designing a new material. Naturally, experimental tests are the gold standard for obtaining this information, but they are costly and time-consuming. The field of material informatics presents an excellent alternative, offering to learn and then predict these properties based on suitable datasets. It is worth mentioning that when mechanical properties are discussed, researchers are dealing with numeric values, leading us to see the mechanical features prediction more as a regression problem. In recent years, applying machine learning approaches to predict material behavior has attracted great attention [40,41,42,43,44].

The prerequisite of material informatics is a trustworthy dataset used as the input for the next steps, such as the one that has been thoroughly explained in the previous sections. This dataset can then be used for quantitative predictions. The next step is identifying features and labels in the dataset. In the context of machine learning, feature refers to the parameters used as the input and label refers to the output corresponding to a set of features [45]. In neural networks, both features and labels must be numeric, meaning that even images are represented by numbers. The act of mapping specific features to labels is called learning, and the choice of how to map these relationships opens the door to learning algorithms [36]. This paper aims to predict three mechanical properties of DP steels, namely UTS, Y, and F, based on PFM-generated microstructures, making them the labels and the feature, respectively.

This research tries to train a hybrid deep-learning model for predicting these mechanical properties based on 1188 PFM-generated microstructure images of DP steel. An overview of the deep learning model is as follows. In this study, the input parameters defined as “labels”, used to train a network for prediction of mechanical properties, are ultimate stress, yield stress and fracture strain for each microstructure. After a deep research study on different transfer learning architectures such as LeNet, Xception, and Inceptionv3 [22] for the material informatics, and having in mind the resemblance of medical images to microstructure images [46], two transfer learning models ResNet50 and VGG16 were trained, and their output was used independently in conjunction with microstructure images to perform deep feature extraction. The Adam optimizer has been implemented as one of the best adaptive gradient-based methods while discussing the optimization functions with the stochastic objective [47]. In order to use it for future estimations, this method saves an exponentially decaying average of previously squared gradients [20]. What makes the Adam optimizer remarkable is the ability to keep the momentum of previous gradients, resulting in a better estimation of the following behavior orders [48]. In addition, it is worth mentioning that Adam’s adaptability to different learning rates is superior and its stability in the convergence process cannot be disputed. This resulted in two feature vectors for each microstructure image, which were then merged to form a stacked feature matrix, which was finally used as the input for the Adaboost and Random Forest (RF) algorithms. Figure 5 illustrates this hybrid deep learning model.

All implementations were performed in Python via the Google Colaboratory platform utilizing an NVIDIA Tesla K80. The Keras, Tensorflow, and SKlearn packages were used to build the final deep network. Training the whole grid with the feature extraction section takes about 2 h, and with access to more advanced hardware and clusters, this could decrease to below one hour. 

### 3.2. VGG16

The model was proposed by Andrew Zisserman and Karen Simonyan at Oxford Visual Geometry Group [49]. Compared to most convolutional neural networks (CNN), the network is simple and works with a simple 3 × 3 stacked layer. VGG is a promising CNN model based on an ImageNet dataset that is trained by over 14 million images to nearly 22,000 categories. To train the model, all images were downsized to 256 × 256. RGB images with a size of 224 × 224 were the inputs for the VGG16 model. Then, the convolutional layers were applied to the images. The whole setup can differ, although the stride, padding, and down-sampling layers can be distinguished. Five max-pooling layers were applied following some of the CNN layers in the very first architecture of the model [50]. The last layer was also equipped with a soft-max layer. An optimum batch size of 16 was selected for the model. It is worth mentioning that the Rectifier Linear Unit (ReLU) function was used in all hidden layers, as depicted below. Other activation functions were also considered. Since we only deal with positive values, ReLU showed the best performance in the case of speed of the convergence, following the mathematical formula in Equation (7) [51]:(7)$$R^{\prime }\left(z\right)=\{\begin{array}{c} z\;\;\;\;\;\;z\geq \;0\;\\ 0\;\;\;\;\;\;z<0 \end{array}.\;$$

### 3.3. ResNet50 (Deep Residual Learning)

Another transfer learning architecture used in the current study was initially designed due to the problematic subject of losing accuracy by adding more layers. The model is called ResNet since it deals with residual learning. The model’s algorithm could justify the superb performance of ResNet in that, instead of modeling the intermediate output, it tries to model the residual of the output of each layer [50]. ResNet50, as the structure has been displayed in Figure 6, is the enhanced model with 48 CNN layers, a max pool, and an average pool. Similar to the VGG16 model, all layers are passed through a ReLU activation function. What matters here is the shortcut connections that skip every three layers in the new ResNet50 model. In comparison to the classic ResNet, every two layers are removed [52], which means that each block of 2 CNN layers in a network of 34 layers was replaced with a bottleneck block of 3 layers. Despite the fact that the ResNet model could be time-consuming, it showed promising performance on the microstructure images. 

### 3.4. Study of Hyper Parameters

Several hyper parameters have different effects on the network, such as learning rate, the number of nodes, dense layers, batch size, or even iteration number. To enhance each model, a set of three hyper parameters, as listed below, is optimized in a deep network with the help of a Keras Tuner, and it will be fixed for the other trials. Until tuning all hyper parameters in a model, this process will go on. The effect of the learning rate, dense layers, and the number of nodes has been investigated and will be discussed in the next section.

The global optimization framework, called Bayesian Optimization [10], is applied to select optimal values. The posterior distribution of this function provides insights into the reliability of the function’s values in the hyper parameter space [53]. With the previously tested values of each iteration, this function tries to take advantage of the variance effect of every defined hyper parameter.

Building a search space [54] for each effective parameter is the main idea behind the Bayesian formulation. With the help of a Keras Tuner in this study, how the performance varies could be detected with the alteration of the values of each hyper parameter. Before applying an automated approach for tuning, a manual grid search was also investigated. Since the process was costly time- and budget-wise, the Keras tuner was a better alternative with the possibility of creating a search space. The same values for three categories of hyper parameters were considered for both models.

#### 3.4.1. Learning Rate (lr)

The learning rate is among the top three in the list of significant hyper parameters in stochastic gradient descent. This factor controls how much the model alters every time weights are updated according to the calculated error of each iteration [55]. Higher learning rates were chosen to accelerate the training at the initial step. Then, lower amounts were applied to avoid any sudden cross-domain fluctuations, especially at the optimal neighborhood. The quantities of lr=(1e−2.1e−3.1e−4.1e−5) were the selected values to test the performance of each model, and the best performance was detected with the implementation of an optimizer. This will be discussed in the Results and Discussion section.

#### 3.4.2. Dense Layers

The most common layer in the ANNs is the dense layer, where the multiplication of matrix vectors occurs. One, two, and three layers were implemented for both VGG16 and ResNet50 networks. Dense units, defined as the output size of each dense layer, were also considered as a hyper parameter. All three models were tested by the change of dense unit numbers. For this study, the range of 16 to 2048 with a step of 32 was considered for tuning the units of the dense layer for both models. The results will be reported in the next part. While discussing the effect of dense layers on the network, the number of layers was also studied. One to three layers were simulated, the most common number of dense layers [56] as one of the most influential parameters in the whole network. The ReLU function for the activation function, which plays the role of neuron transformation for each layer, was designated.

#### 3.4.3. Regression Ensemble Learning Method

Keeping in mind that the regression part of the model could also be a turning point in the simulation, two main methods based on the decision tree algorithm were nominated for the learning method in the last part of the model to predict the mechanical properties. Adaboost and Random Forest architectures are illustrated in Figure 7. In the first method, which is quite a famous method called Random Forest (RF), every decision tree for the optimal splits takes a random volume of features according to the bagging method, meaning that each tree is trained individually with a random subset of data but with equal weights. However, in the following method, which we are focused on, called Adaboost, each tree takes its own weight by analyzing the mistakes of the previous one and increases the weight of misclassified data points, which is called the boosting method. The ordering of the trees in the Adaboost method could affect the subsequent layers, although each tree performs independently with RF. The algorithms of both models are sketched in the demo below.

### 3.5. Models’ Performance Analysis

Different types of errors could be considered to have a mutual understanding of the performance of a model. Different methods were considered to analyze the model’s performance and calculation of accuracy. To visualize each model’s performance, training loss and validation loss, abbreviated as “loss” and “val_loss”, respectively, were calculated as evaluation measures according to mean square error (MSE). Training loss, formulated with the cost function, is the measurement that is calculated after each batch, depicting how accurately a deep model can fit the training dataset. In contrast, validation loss is defined for assessing the performance of the dataset that was put aside. Root mean square error (*RMSE*) is the second approach for monitoring the error in the current study, reported in some studies as the error that can outperform some other types, such as weighted MSE [20]:(8)$$MSE=\frac{1}{N}\sum_{i}^{N}{\left(y_{i}-{\hat{y}}_{i}\right)}^{2},\;$$
(9)$$RMSE=\sqrt{\frac{1}{N}\sum_{i}^{N}{\left(y_{i}-{\hat{y}}_{i}\right)}^{2}},\;$$
where $y_{i}$, ${\hat{y}}_{i}$, and N are the ground truth practical values, the predicted effective values of the mechanical property (UTS, Y, and F in this study), and the number of selected data in a set for which error is calculated, respectively. In addition, $y_{average}$ is the average value of the aforementioned property in the dataset [13].

As we are dealing with a regression algorithm, two mathematical errors are also calculated during the prediction of mechanical properties, enabling us to have a perspective of how to compare the final results of the regression step. For both categories of training and testing datasets, mean absolute error (*MAE*) and mean absolute scaled error (*MASE*) (Ref. [57]) are estimated as:(10)$$MAE=\frac{1}{N}\sum_{i}^{N}\left|y_{i}-{\hat{y}}_{i}\right|,$$
(11)$$RMASE=\frac{1}{N}\sum_{i}^{N}\left|\frac{y_{i}-{\hat{y}}_{i}}{y_{average}}\right|\times 100\%,$$

## 4. Results and Discussion

Some traditional approaches are presented in different micromechanical studies, though not every microstructure output after augmentation can lead to the same mechanical properties, which seems to be ignored in some studies. Flipping up to down, abbreviated as Flip UD, flipping left to right (LR), and rotating clockwise (CC) and counterclockwise (CCW) approaches were investigated, respectively, and each time, about 2000 to 4000 images were produced. Additionally, according to some studies [58] with the shear approach for data generation, this method with −16 < shear angle < +16 also was studied. 

This could be significant evidence of how microstructure investigation is a criterion that needs more consideration while being on data generation. In many datasets, machine-learning networks could be problem solvers, for either classification or regression problems, and the criteria for boosting the dataset Table 6 still shows a good amount of error values; however, this study’s primary dataset had better results without boosting with traditional methods. The presence of two phases of ferrite and martensite and, more importantly, the interface of phases means that methods such as cropping could change the mechanical properties. Among them, flipping is the method with a minimum error of 2 percent; however, it is still not as low as the genuine dataset itself with no augmentation.

The performance of the two models individually, as discussed in the previous section, is reported in Figure 8, demonstrating the decreasing trends of performance evaluation errors. As discussed in studies including the deep learning keywords, if validation loss shows a big difference in value to training loss, overfitting occurs, which did not happen in this model, as can be evidently seen in the diagram below. To avoid the occurrence of under-fitting, each epoch was carefully monitored with the illustration of MSE for both training data and validation. The ResNet50 model shows smoother behavior, while the VGG16 model results in fewer error values sooner. The analysis of both models and other deep learning approaches emphasizes the advantages of both models in combination.

With a fixed batch size of 16 for both models, the results after the optimization are depicted below. While it has been reported by [59] that 32 is a better value for the batch size, before the optimization process for other values, it was studied manually with different iterations, and in this case, a batch size of 16 showed better performance, as was also reported in some other studies [60]. The details of the parametric study are listed in Table 7. The epochs represent each time the model and the whole training dataset were trained. The batches were picked randomly for each epoch, and the testing dataset (10 percent of the dataset) was randomly separated to see the model validation performance. This must be done to avoid further overfitting while running the subsequent simulations. On 52 simulations, each epoch was performed, and the number of epochs was fixed to 200 and manually optimized. We might obtain better results with more epochs, though overfitting is more probable and time-consuming. The HP optimization section of the ResNet50 model could take less than 40 min, and the VGG16 optimization could take less than an hour.

Regarding the timing issue and monitoring the running epochs, it needs to be pointed out that the training time since Callback defined in the model could be less than 15 min. Callbacks in the Keras library help in periodically saving the model and monitoring the metrics after each batch, and EarlyStopping Callback could stop training after it is triggered. Along with the ModelCheckpoint Callback, the best model while training the network with a defined measurement factor for the validation data performance could be saved. 

MASE, as the main error for the overview of each simulation, is reported in Table 8. Adaboost and RF errors are also reported for the three mechanical properties in this study. According to the authors’ knowledge, this is the first study that gives an optimization performance analysis to investigate the effect of more than three hyper parameters. Table 8 tries to report the errors for the ResNet50 optimized trial, which showed acceptable performance for the prediction of ultimate and yield stress with about 3 percent MASE error. The other model’s performance was also investigated, and the optimized trial errors were reported, respectively. Unfortunately, VGG16, as reported the results in the Table 9, with an approximate error of 15 and 11 percent for ultimate and yield stresses, respectively, could not perform more accurately.

It is worth noting that the two models could reasonably identify stress characteristics. However, in the case of reporting the strain performance, they could not correspond better than 10 percent with the testing samples. This could result from the scarcity of data that is needed when it comes to fracture strain investigation, such as crack initiation or crack propagation pattern. The dependence of the amount of strain on morphology is more than that of stress values, which could also explain why better results were obtained in the case of stress investigation [2]. 

The optimized model proved its best performance with an error of 1.3% for the testing dataset for the prediction of ultimate stress, 0.9% for the yield stress, and 6% for the prediction of fracture strain of the same data (stated in the following table for each parameter).

The errors for each model for both regression approaches are listed in Table 10. It is worth noting that while each model works individually, the hybrid model outperforms them. The results in every step are compared to the ground truth and are depicted in Figure 9 and Figure 10 for both applied regressors, Adaboost and RF, respectively.

As it is evident in the fluctuation plots in Figure 9 and Figure 11. The hybrid model, after running the feature extraction [61] with the adaptively fusing loss function as discussed in [62], could grasp most of the fluctuations and predict the values of peaks and bottoms better.

As discussed for each model in previous section, the same characteristics could be detected in the hybrid model while talking about the fracture strain. The parity plots in Figure 11 and Figure 12 could be good representatives of how the prediction of strain behavior could still be challenging. Even though each model benefits from less time-consuming simulations, running the feature extraction section in the hybrid model could satisfy us with the difference in the outputs for the testing trials. Considering the model’s excellent performance in analyzing all three mechanical properties, we can ignore the fact that it could be tedious. Neither the testing dataset nor the training set shows significant deviation to either side of the ideal regression line (r = 1).

Last but not least is this model’s outstanding performance of RF regression. More evidently, in the scatter plots, this could be recognized. Machine learning at this moment proves its significant role in eliminating costly experiments, especially regarding topics such as material characteristics which, for a substantial amount of time, could only be validated against experiments.

## 5. Conclusions

In the presented work, a dataset of microstructures using Phase Field Modeling based on the experimental microstructures was created, containing 1188 RVEs of dual-phase steel in different heat treatment conditions. The FEA technique labeled the entire dataset with three mechanical properties. This aims to feed a deep learning approach that was implemented with the help of two transfer learning approaches, VGG16 and ResNet50, called the hybrid model. Before building the final model, a parametric study was performed to optimize each model to access the best features of both VGG16 and ResNet50 models. Moreover, a comparison of decreasing trends of performance evaluation errors between the two models was also explored. The results show that with the implementation of tuning, which leads to the optimization of hyper parameters, each model independently could not show a fair prediction of mechanical properties. In contrast, the hybrid model claims to predict the mechanical properties, including the ultimate and yield stresses, with less than two percent of mean absolute squared error (MASE). However, in the case of fracture strain, some challenges still exist due to the relationship between this parameter and the morphology, which is more than that of stress values. 

To optimize the model, we used three dense layers for the ResNet50 model with 992, 671 and 32 nodes, and one layer for the VGG16 model with 992 nodes. The learning rate effect was investigated and optimum values of 1 × 10^−4^ for VGG16 and 1 × 10^−3^ for ResNet50 were nominated. The number of convolutional layers was the third hyper parameter that we focused on and the best number of filters was 16 for the VGG16 model and 16 for the ResNet50 model, respectively. 

Two regressors’ performances (Random Forest (RF) and Adaboost) were also observed, and at each evaluation, the RF showed promising results, with about 1 to 3 percent of enhanced error. Data augmentation was also investigated carefully to determine how approaches such as flipping, rotation, or shearing could boost the dataset numerically but not the results of the prediction step. Finally, we proposed an optimized model with a great accuracy of 98% to predict the mechanical properties, which is an excellent demonstration of how applicable ANN could be in the field of material informatics. 

## Figures and Tables

**Figure 1 materials-16-00447-f001:**
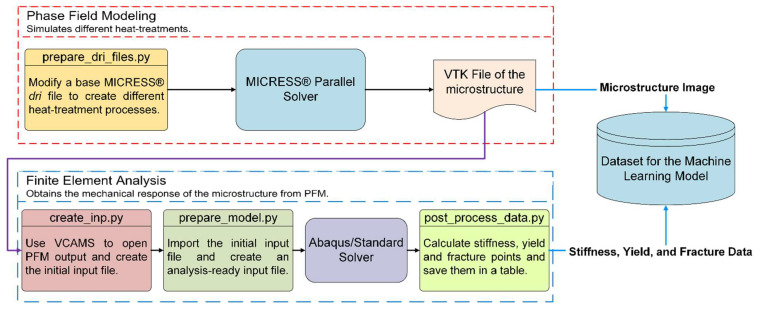
Workflow of the microstructure data generation with different heat treatment conditions.

**Figure 2 materials-16-00447-f002:**
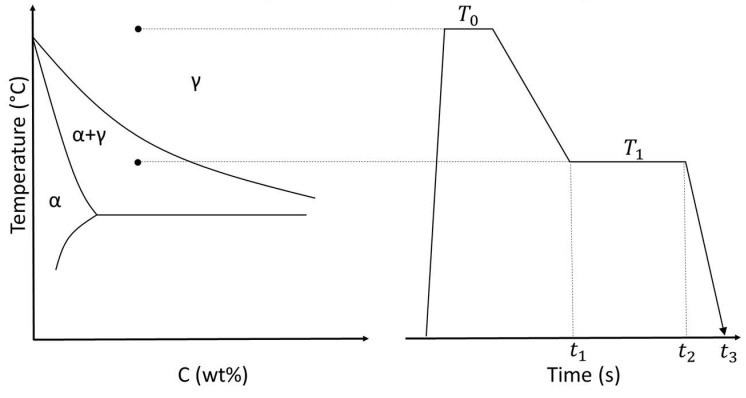
Schematic view of the step-quenching heat treatment process routine.

**Figure 3 materials-16-00447-f003:**
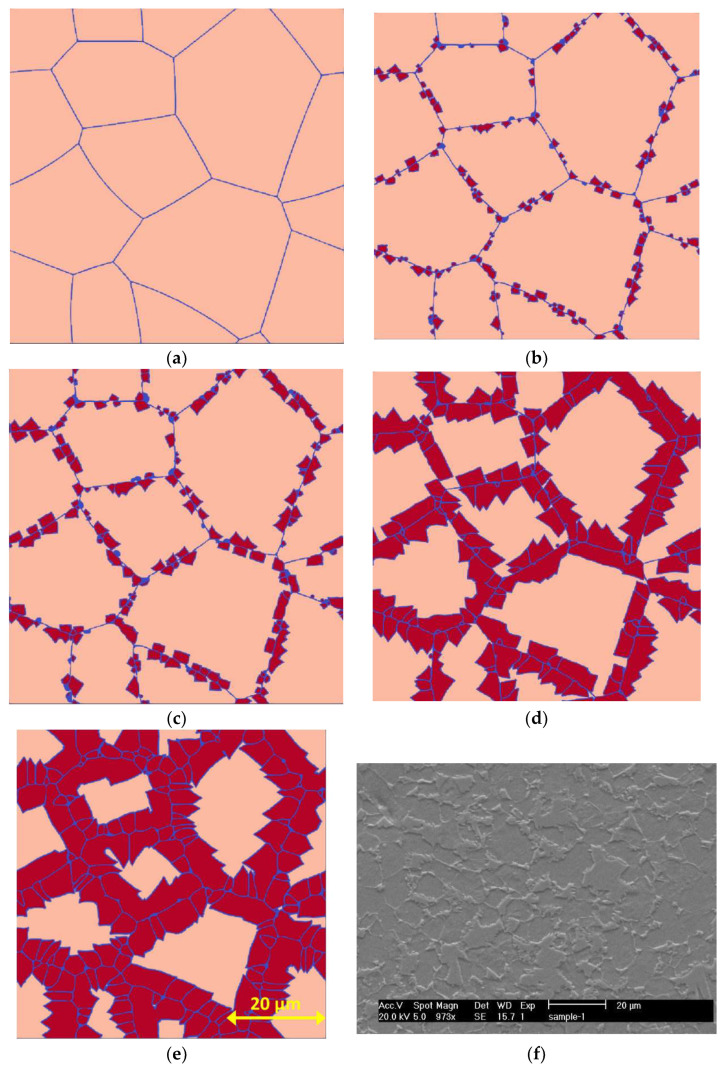
Progression of the results of the PF simulation: (**a**) initial state, (**b**) 15 s, (**c**) 1 min, (**d**) 10 min, and (**e**) after quenching; and (**f**) SEM image of a sample undergoing the same heat treatment.

**Figure 4 materials-16-00447-f004:**
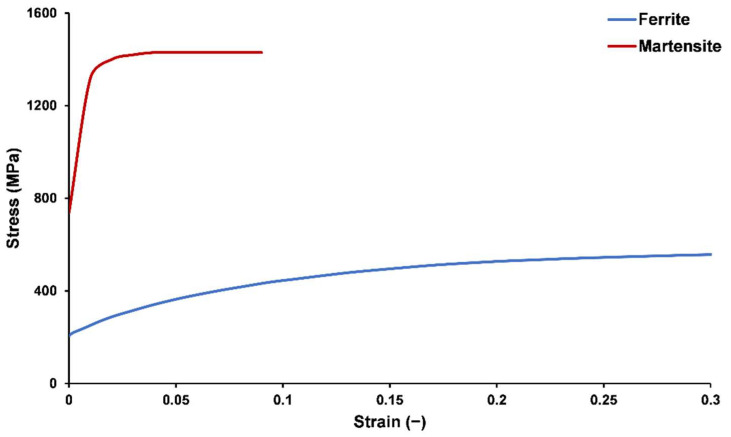
Flow curves of the ferrite and martensite phases used in micromechanical FEA [2].

**Figure 5 materials-16-00447-f005:**
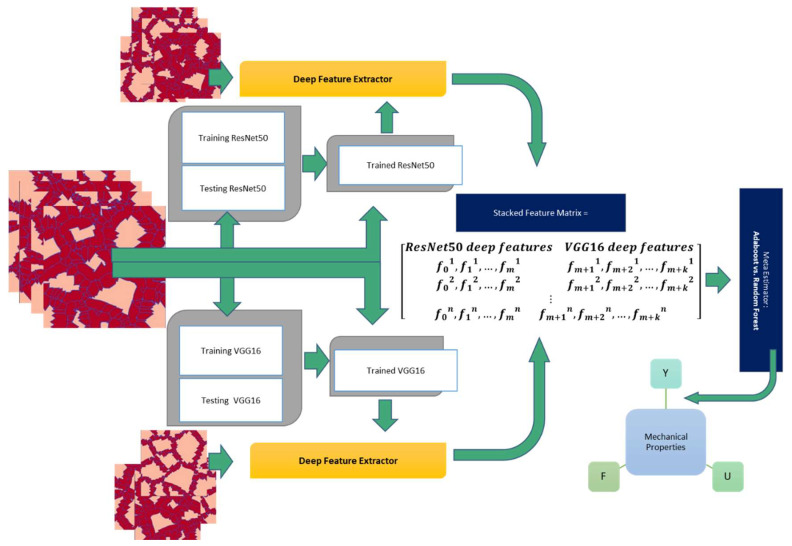
General framework of the hybrid model.

**Figure 6 materials-16-00447-f006:**
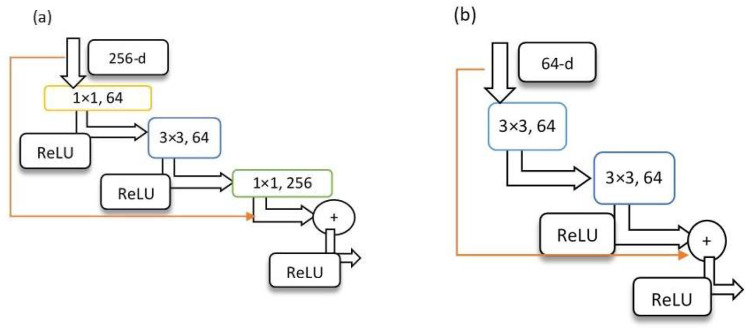
(**a**) ResNet50 model vs. (**b**) ResNet classic model.

**Figure 7 materials-16-00447-f007:**
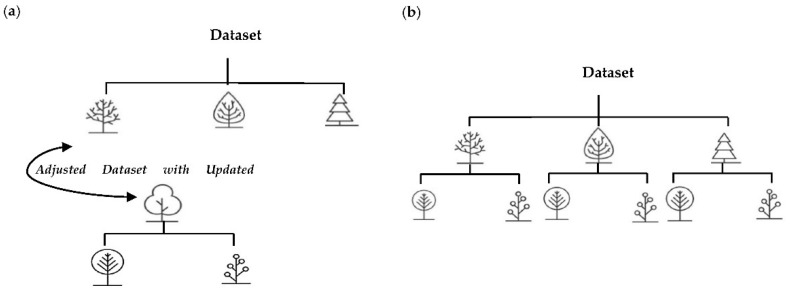
(**a**) Decision stumps in Adaboost based on the boosting method; (**b**) decision trees in RF with the bagging method.

**Figure 8 materials-16-00447-f008:**
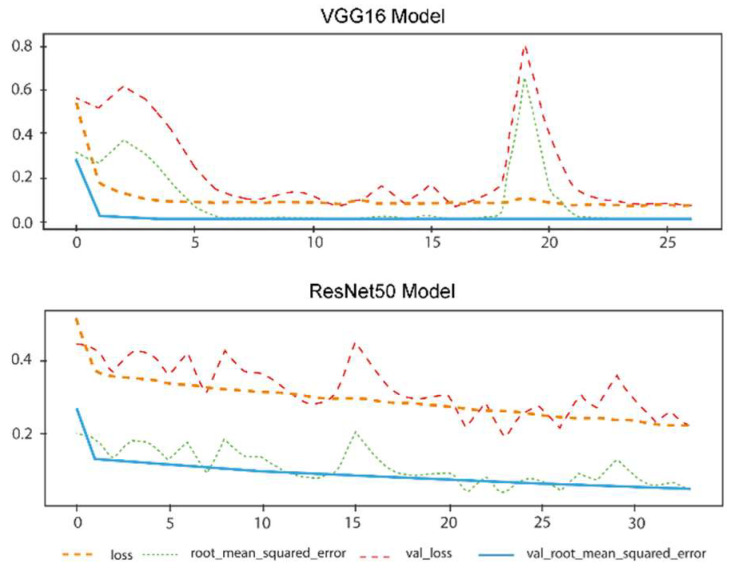
Loss and validation loss diagram for two approaches, ResNet50 and VGG16.

**Figure 9 materials-16-00447-f009:**
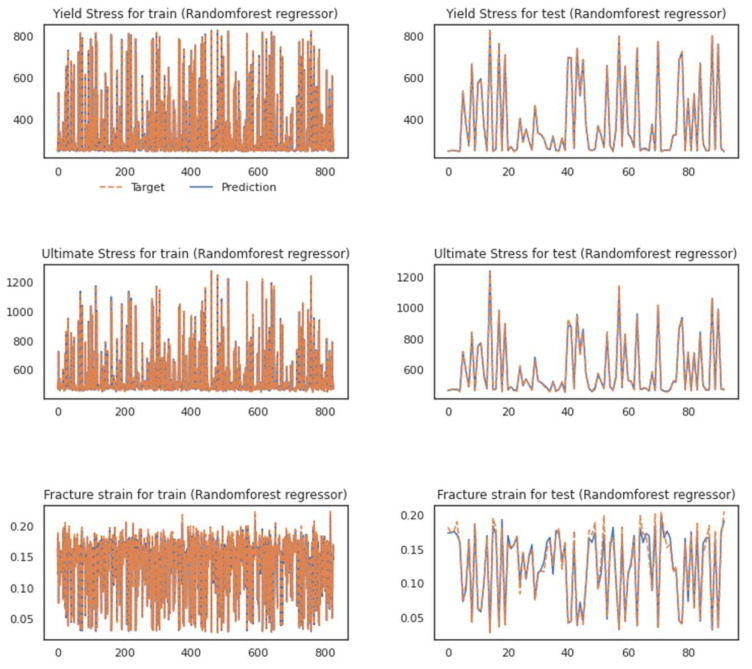
Performance of the proposed hybrid model while using the RF regressor for the training set (left diagrams) and test set (right figures).

**Figure 10 materials-16-00447-f010:**
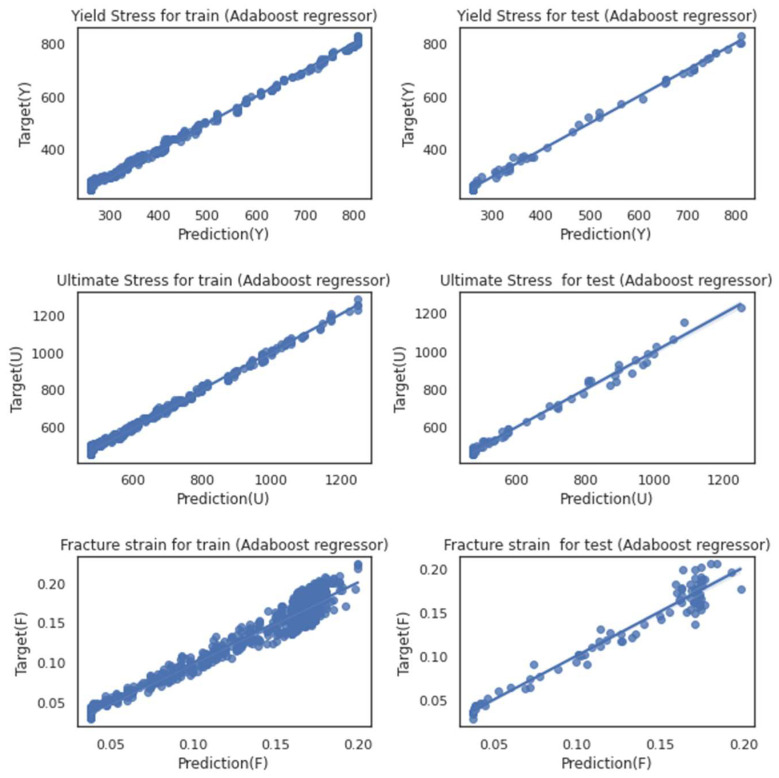
Dataset parity plot while using the Adaboost regressor for training (right figures) and testing (left figures) datasets for three mechanical properties: yield stress (Y), ultimate stress (U), fracture strain (F).

**Figure 11 materials-16-00447-f011:**
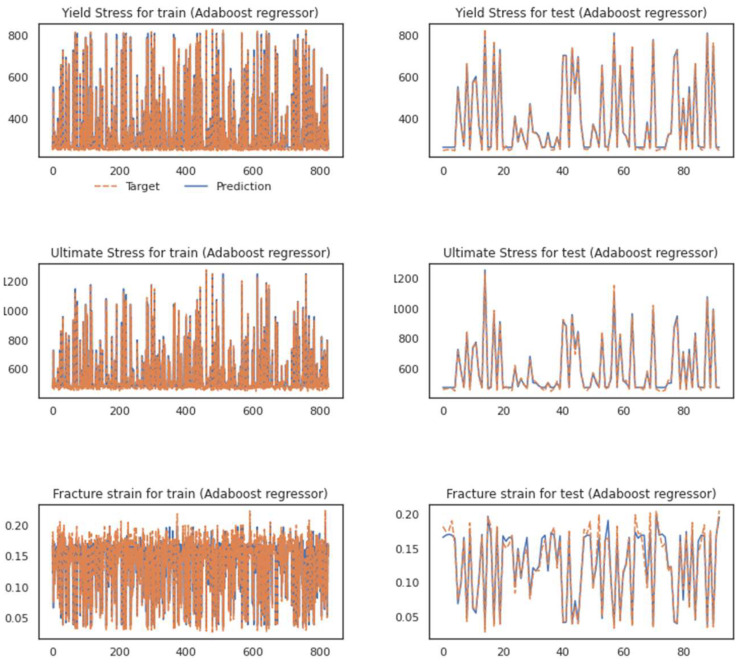
Performance of the proposed hybrid model while using the Adaboost regressor.

**Figure 12 materials-16-00447-f012:**
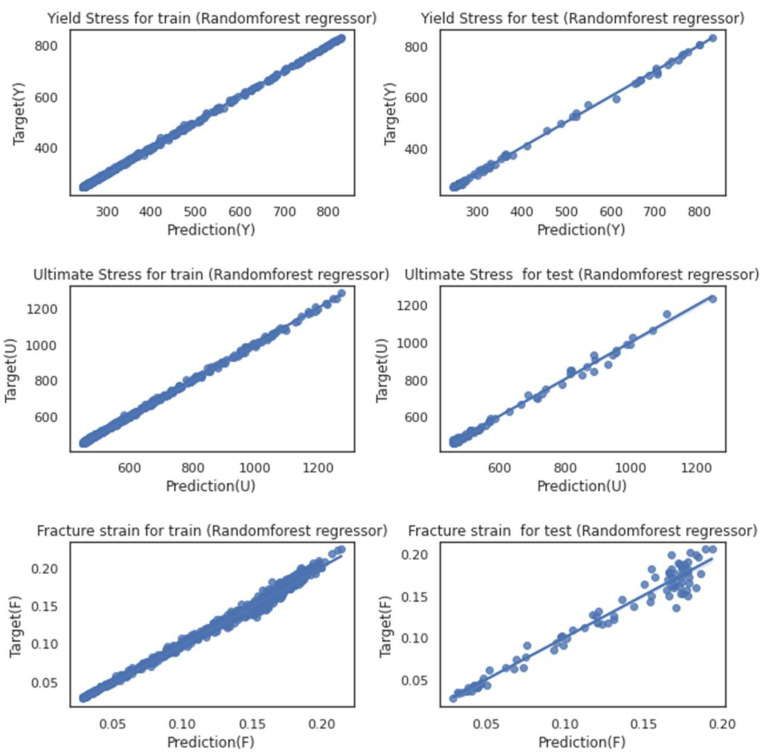
Scatter diagram of training and testing microstructure images while using the RF regressor for training (right figures) and testing (left figures) datasets for three mechanical properties: yield stress (Y), ultimate stress (U), fracture strain (F).

**Table 1 materials-16-00447-t001:** Chemical composition of the low-carbon steel used for validating the PF model.

Element	C	Mn	Si	P	S	Cr	Mo	V	Cu	Co
**wt%**	0.2	1.1	0.22	0.004	0.02	0.157	0.04	0.008	0.121	0.019

**Table 2 materials-16-00447-t002:** Linearized data for the phase diagram at $T_{1}=1043$.

	Phase Boundary	α/γ + α	γ/α + γ
**Carbon (** $C_{ij}$ **)**	**Concentration (wt%)**	0.0048	0.365
	**Slope (°K/wt%)**	−13,972.00	−188.80
**Manganese (** $Mn_{ij}$ **)**	**Concentration (wt%)**	1.58	3.78
	**Slope (°K/wt%)**	−100.03	−23.55
	$L_{ij}$ ** $\left(J\;cm^{-3}\right)$ **	0.238	

**Table 3 materials-16-00447-t003:** Interfacial parameters between ferrite (α) and austenite (γ) [3,32].

Interface	α/α	α/γ	γ/γ
**Interfacial energy (J** $cm^{-2}$ **)**	7.60 × $10^{-5}$	7.20 × $10^{-5}$	7.60 × $10^{-5}$
**Mobility (** $cm^{4}J^{-1}s^{-1}$ **)**	5.00 × $10^{-6}$	2.40 × $10^{-4}$	3.50 × $10^{-6}$

**Table 4 materials-16-00447-t004:** Comparison of mechanical behavior and experimental results.

	Yield Stress (MPa)	Ultimate Stress (MPa)	Fracture Strain (−)
**Numerical**	314.36	517.9	0.127
**Experimental**	323.7	530.1	0.131

**Table 5 materials-16-00447-t005:** Heat treatment parameters and their values. The units for temperatures, times, and cooling rates are Kelvin, seconds, and $\frac{K}{s}$, respectively.

Parameter	Description	Values
$T_{0}$	Initial temperature of the microstructure.	1250
$CR_{01}$	Cooling rate between points 0 and 1. Not used directly.	−10, −5, −1
$t_{01}$	Number of seconds it takes to cool down from point 1 to point 2.	Calculated based on $CR_{01}$
$T_{1}$	Temperature of the microstructure in point 1.	1000, 1010, 1020, 1030, 1040, 1050, 1060, 1070, 1080, 1090, 1100, 1110
$t_{12}$	Holding time between points 1 and 2 in seconds.	10, 20, 30, 60, 300, 600, 900, 1800, 3600, 7200, 10,800
$T_{2}$	Temperature of the microstructure in point 2.	Equal to $T_{1}$
$CR_{23}$	Cooling rate between points 2 and 3 based on the quench media. Not used directly.	Brine=−220 Water=−130 Oil=−50
$t_{23}$	Number of seconds it takes to cool down from point 1 to point 2.	Calculated based on QM
$T_{3}$	Room temperature.	298

**Table 6 materials-16-00447-t006:** MASE comparison for three labels of mechanical properties with different methods of traditional augmentation, such as flipping, rotating, and shearing.

**Rotate 90 CC**		**MASE Y**	**MASE U**	**MASE F**
**Hybrid Model Error Report (%)**	Train	10.196	6.681	11.215
	Test	10.209	6.308	11.440
Rotate 90 CCW		**MASE Y**	**MASE U**	**MASE F**
**Hybrid Model Error Report (%)**	Train	10.89	6.502	12.002
	Test	11.01	6.401	11.928
Random Shear		**MASE Y**	**MASE U**	**MASE F**
**Hybrid Model Error Report (%)**	Train	1.232	0.943	2.9172
	Test	3.534	2.346	8.134
Flip UD		**MASE Y**	**MASE U**	**MASE F**
**Hybrid Model Error Report (%)**	Train	1.4474	1.0802	3.0451
	Test	4.623	2.971	8.677
Flip LR		**MASE Y**	**MASE U**	**MASE F**
**Hybrid Model Error Report (%)**	Train	1.0306	0.8930	2.744
	Test	2.692	2.195	7.025

**Table 7 materials-16-00447-t007:** Parameters choice list for the optimization of two methods of transfer learning, VGG16 and ResNet50.

Parameter	Description	Values	VGG16 Optimized Values	ResNet50 Optimized Values
E	Epoch numbers	200	200	200
lr	Learning rate	1 × 10^−2^, 1 × 10^−3^, 1 × 10^−4^, 1 × 10^−5^	1 × 10^−4^	1 × 10^−3^
Conv2D	Number of filters in the convolution layer	min = 16, max = 512, step = 32	336	16
Dense Units (layer1)	Output size of each dense layer	min = 32, max = 1024, step = 64	992	992
Dense Units (layer2)			-	672
Dense Units (layer3)			-	32

**Table 8 materials-16-00447-t008:** ResNet50 error report for training and testing data after the HP optimization.

		MASE Y	MASE U	MASE F
**Resnet50 Model Error Report (%)**	Train	5.559	3.713	8.092
	Test	5.465	3.610	10.448

**Table 9 materials-16-00447-t009:** VGG16 error report for training and testing data after the HP optimization.

		MASE Y	MASE U	MASE F
**VGG16 Model Error Report (%)**	Train	13.043	16.071	36.67
	Test	11.292	15.001	41.963

**Table 10 materials-16-00447-t010:** Hybrid model error report for training and testing data after the HP optimization, while considering two approaches, Adaboost and RF regressors.

Hybrid Model Error Report		MASE Y	MASE U	MASE F
**Adaboost (%)**	*Train*	2.532	1.625	6.323
	*Test*	2.387	2.172	6.881
		**MASE Y**	**MASE U**	**MASE F**
**Random Forest (%)**	*Train*	0.386	0.494	2.432
	*Test*	0.924	0.574	6.670

## Data Availability

The data set of this study was generated at “Institut für Materialprüfung, Werkstoffkunde und Festigkeitslehre” (IMWF). Data could be made available upon reasonable request under the supervision of the IMWF institute. The proposed model is available upon reasonable request to the corresponding author.

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
