# Peer review of "Hybrid Data-Driven Deep Learning Framework for Material Mechanical Properties Prediction with the Focus on Dual-Phase Steel Microstructures"

_materials, 2023, doi:10.3390/ma16010447_

Round 1
Reviewer 1 Report
In this study, the authors created a reliable data pipeline consisting of experimentally-validated phase field simulations and finite element analysis, to generate a dataset of dual-phase steel microstructures and mechanical behaviors under different heat treatment conditions. And then, the authors presented a deep learning-based method, which was the hybridization of two well-known transfer-learning approaches, ResNet50 and VGG16. By fusing the hybrid model and feature extractor, the dual-phase steels' yield stress, ultimate stress, and fracture strain under new treatment conditions were predicted with an error of less than 1%. This work is comprehensive, with some advances in the application of the method.
Totally speaking, this topic is interesting in this field and the paper is well organized. Therefore, it is in the opinion of this reviewer that the paper may be accepted for publication provided that the following minor revisions are made.
1. First of all, the novelty in this paper is somewhat unclear. The authors may further strengthen the description for the novelty in this study at the end of the Introduction. Does this novelty come from the problem under consideration, or from the method used?
2. In Abstract, the sentence at lines 16-18 is hard to understand, please check the grammar.
3. The quality of Figure 5 should be improved further, since the texts in this figure are not clear and difficult to identify.
4. After line 115, Equation (2) should be Equation (3).
5. Some text arrangement errors and typos should be found and rectified, for example, at lines 102, 107, 382 and 387, after line 301, to list but a few.
6. The style of the References did not meet the requirements of the journal, for example, all authors’ names should be given; there is no need to provide the publication month, etc. Please refer to the template file for this journal Materials.
Reviewer 2 Report
The manuscript is interesting. However, there are several remarks.
1. I couldnot find full information about input parameters of the model.
2. During phase transformations taking place near the lattice defects the activation energy is less then the bulk diffusion activation energy, which is given in the Table 4 (see, for example, Kaputkin D.E. Correlation between the thermokinetic parameters of diffusional decomposition and the activation energy of diffusion in steels and nonferrous alloys. Physics of Metals and Metallography, 2005, v.99, #4 (April), pp. 343 - 347).
3. Simutaneous application of Celsius degree (e.g., p. 2.2.2) and Kelvin (e.g. Table 6) is not reasonable.
4. Phase boundaries mentioned in the Table 2 are alpha / (alpha + gamma) and gamma / (alpha + gamma) but not alpha/gamma and gamma/alpha.
